# Analysis of Flavonoid Metabolites in Watercress (*Nasturtium officinale R. Br.*) and the Non-Heading Chinese Cabbage (*Brassica rapa* ssp. *chinensis cv. Aijiaohuang*) Using UHPLC-ESI-MS/MS

**DOI:** 10.3390/molecules26195825

**Published:** 2021-09-26

**Authors:** Xiaoqing Ma, Qiang Ding, Xilin Hou, Xiong You

**Affiliations:** 1State Key Laboratory of Crop Genetics & Germplasm Enhancement, Key Laboratory of Biology and Genetic Improvement of Horticultural Crops (East China), Ministry of Agriculture and Rural Affairs of the P. R. China, Engineering Research Center of Germplasm Enhancement and Utilization of Horticultural Crops, Ministry of Education of the P. R. China, Nanjing Suman Plasma Engineering Research Institute, Nanjing Agricultural University, Nanjing 210095, China; 2018204021@njau.edu.cn (X.M.); j23_xf@163.com (Q.D.); 2College of Sciences, Nanjing Agricultural University, Nanjing 210095, China

**Keywords:** *Nasturtium officinale R. Br.*, *Brassica rapa* ssp. *chinensis cv. Aijiaohuang*, flavonoid metabolites, PCA, OPLS-DA, UHPLC-ESI-MS/MS

## Abstract

Flavonoids from plants play an important role in our diet. Watercress is a special plant that is rich in flavonoids. In this study, four important watercress varieties were compared with non-heading Chinese cabbage by ultra-high-performance liquid chromatography-electrospray ionization-tandem mass spectrometry (UHPLC-ESI-MS/MS). A total of 132 flavonoid metabolites (including 8 anthocyanins, 2 dihydroflavone, 3 dihydroflavonol, 1 flavanols, 22 flavones, 11 flavonoid carbonosides, 82 flavonols, and 3 isoflavones) were detected. Flavonoid metabolites varied widely in different samples. Both the non-heading Chinese cabbage and the variety of watercress from Guangdong, China, had their own unique metabolites. This work is helpful to better understand flavonoid metabolites between the non-heading Chinese cabbage and the other four watercress varieties, and to provide a reliable reference value for further research.

## 1. Introduction

Jennifer Di Noia defined powerhouse fruits and vegetables in 2014. Of 47 foods studied, the nutrient density score of watercress is the highest, followed by Chinese cabbage [1]. Watercress is rich in flavonols [2] and vitamins B1, B2, and E [1,3]. Increased vegetable intake, particularly of *Brassicaceae* vegetables such as watercress, cabbage, broccoli, cauliflower, mustard greens, and brussel sprouts, has been linked to a reduced risk of several types of cancer in human population studies [4,5,6,7,8,9,10]. Watercress has been shown in many past studies to have beneficial effects in humans, such as treating inflammation, chemopreventive benefits, and so on. Several studies have shown that watercress extract can inhibit the growth and metastasis of cancer cells in vitro [11,12,13,14,15]. *Brassicaceae* vegetables, particularly cauliflower, broccoli, cabbage, and watercress, have been shown to reduce oxidative DNA damage in in vitro experimentation in human cells [16].

Metabolomics is the comprehensive qualitative and quantitative analysis of all small molecules in a cell, tissue, or organism to study the interaction of internal and external factors. As a bridge between plant genotype and phenotype, metabolites play an important role in plant growth and development [17]. The most common analytical methods were used for multiple chemical compounds’ determination in plants such as gas chromatography (GC) coupled with MS/MS (GC–MS), high-performance liquid chromatography (HPLC) coupled with fluorescence detection or ultraviolet (UV), and immunological approaches (ELISA) [18,19]. However, these methods showed many shortcomings and technical defects, such as time-consuming sample running, peak interferences, poor sensitivity, and shortness of strong chromophores. In recent years, widely targeted metabolites based on UHPLC-ESI-MS/MS have been widely used owing to its high throughput, super sensitivity, wide coverage, and qualitative and quantitative accuracy [20,21]. Compared with traditional HPLC, UHPLC-MS has the advantages of high separation efficiency, short analysis time, and less solvent consumption. In addition, it can obtain a more comprehensive chemical profile and quantification by utilizing different ion modes and having higher sensitivity. Based on the previously constructed MS2T library and the widely targeted profiling method, we are able to screen a large number of samples for the quantitation of metabolites [22]. Further, it is now a very powerful tool to gain a thorough understanding of plant metabolism.

In this study, the UHPLC-ESI-MS/MS metabolomics method was used to evaluate the difference of metabolites between the non-heading Chinese cabbage and watercress. This is the first attempt to study the metabolomics between non-heading Chinese cabbage and watercress and between different varieties of watercress. In this study, five *Brassicaceae* cultivars, including four watercress varieties (MG, WH, GD, and YD) and one non-heading Chinese cabbage (AJH), were selected as experimental materials. The purpose of this experiment was to study the differences of flavonoid metabolites between watercress and Chinese cabbage as well as those among different cultivars of watercress. This study provides a theoretical basis for further study of flavonoid metabolites between the non-heading Chinese cabbage and four watercress varieties and provides a reference for making full use of them in the future.

## 2. Materials and Methods

### 2.1. Experimental Materials and Methods

Non-heading Chinese cabbage (*Brassica rapa* ssp. *chinensis cv. Aijiaohuang*) (AJH) and four common watercress varieties were selected as experimental varieties. Seeds of the non-heading Chinese cabbage were soaked for germination and grown in pots containing a soil/sand mixture (3:1) in a controlled artificial climatic chamber (16 h light/8 h dark photoperiod at 22 °C/18 °C) at Nanjing Agricultural University. Four varieties of watercress were from United States (MG), Wuhan China (WH), Guangdong China (GD), and Yangzhou China (YD), respectively. They were also grown in a growth chamber under the same conditions. After 40 days of cutting planting, the leaves were collected from plants. The sample was then placed into liquid nitrogen immediately after collection.

### 2.2. Determination of Total Flavonoids Content

Approximately 2.0 g of leaves of each variety was taken and dried at 65 ℃ for 3 days to a constant weight. Then, it was ground into powder and 0.02 g was taken to be measured. Total flavonoids content was determined according to the protocol of Plant Flavonoids Test kit (Suzhou Comin Biotechnology Co., Ltd., Suzhou, China).

### 2.3. Extraction Process

The freeze-dried samples were crushed by a mixed mill (MM 400, Retsch) with zirconia beads at 30 Hz for 1.5 min. Further, 100 mg powder was weighed and extracted overnight at 4 ℃ with 1.2 ml 70% aqueous methanol. Before UHPLC-MS/MS analysis, 12,000 rpm was centrifuged for 10 min for filtration. Further, the list of reagents with purities and manufacturers is shown in Appendix A.

### 2.4. Chromatographic and Mass Spectrometry of Analysis Conditions

Ultra-high-performance liquid chromatography (UHPLC) (SHIMADZU Nexera X2, https://www.shimadzu.com.cn (accessed on 10 January 2020)) and tandem mass spectrometry (MS/MS) (Applied Biosystems 4500 QTRAP, http://www.appliedbiosystems.com.cn/ (accessed on 10 January 2020)) were the main systems used in the detection. The analysis conditions are shown in Table 1. Then, the effluent was alternatively connected to an ESI-triple quadrupole-linear ion trap (QTRAP)-MS. The main contents of mass spectrum conditions are shown in Table 2. In triple quadrupole (QQQ), each ion pair is scanned and detected according to the optimized declustering potential (DP) and collision energy (CE) [22].

### 2.5. Qualitative and Quantitative Principles of Metabolites

According to the self-built database MWDB (Metware Biotechnology Co., Ltd. Wuhan, China), the qualitative analysis of the secondary spectrum information data was performed. Isotopic signals; duplicate signals containing K^+^, Na^+^, and NH4^+^ ions; and fragments of other larger molecular weight substances were removed from the analysis. Multiple reaction monitoring (MRM) analysis of QQQ mass spectrometry was used to perform the quantitative analysis of metabolites. After obtaining the metabolite spectrum analysis data of different samples, the peak area integral was performed for all mass spectra, and the peak of the same metabolite in different samples was integrated and corrected.

### 2.6. Statistical Analysis

R (http://www.r-project.org/ (accessed on 10 February 2020)) was used for cluster analysis, as well as PCA and OPLS-DA in accordance with previously described methods [23]. Differential metabolism interacts in the organism and forms different pathways. Differential metabolites were annotated and demonstrated using the Kyoto Encyclopedia of Genes and Genomes (KEGG) database.

## 3. Results

### 3.1. Determination of Total Flavonoid Content

The content of total flavonoids from four watercress cultivars and a non-heading Chinese cabbage cultivar was determined in the study, including the watercress cultivars from the United States (MG), Wuhan China (WH), Guangdong China (GD), Yangzhou China (YD), and the non-heading Chinese cabbage (*Brassica rapa* ssp. *chinensis cv. Aijiaohuang*). The flavonoid content of GD was the highest at 14.0 mg/g (Figure 1). The flavonoid contents of all four watercress varieties were higher than that of the non-heading Chinese cabbage.

### 3.2. Qualitative and Quantitative Analyses of Metabolites and Quality Control (QC) Analysis of Samples

The typical total ions current (TIC) plot represents a continuous map obtained by adding the intensity of all ions in the mass spectrum at each time point. Figure 2A shows a typical TIC plot of one QC sample. The metabolites detected were analyzed qualitatively and quantitatively based on the local metabolic database. Figure 2B shows the multi-peak diagram of metabolite detection in the MRM model. This result shows what can be detected in the sample, with different color mass spectrum peaks representing different metabolites. The x-coordinate represents the retention time (RT) of the metabolites and the y-coordinate represents the ion current intensity of ion detection.

A total of 132 flavonoid metabolites were identified, including 8 anthocyanins, 2 dihydroflavone, 3 dihydroflavonol, 1 flavanols, 22 flavones, 11 flavonoid carbonosides, 82 flavonols, and 3 isoflavones. Details of each metabolite are shown in Appendix A, such as the metabolite name, number, and peak integral value.

The QC sample is a mixture of sample extracts to analyze the repeatability of the sample under the same treatment method. During instrumental analysis, one quality control sample was inserted into every 10 test and analysis samples to monitor the repeatability of the analysis process. The reproducibility of metabolite extraction and detection could be determined by overlapping display and analysis of the TIC diagrams of different QC samples with essential spectrum detection and analysis.

Appendix A showed an overlay of the TIC plots, and the results show that the peak diagram of metabolites has a high degree of overlap. The repeatability of metabolite extraction and detection can be determined by the consistency of retention time and peak strength.

### 3.3. Principal Component Analysis (PCA) for the Different Varieties

Through PCA of samples, the differences of total metabolism between samples of each group and the degree of variation among samples within the group can be preliminarily understood. In this study, PC1 and PC2 were extracted, which were 48.43% and 24.33%, respectively. The cumulative contribution rate reached 72.76%. The PCA score plot showed that AJH, MG, WH, YD, and GD were clearly separated, and the repeated samples were compactly collected together (Figure 3). The results indicated the repeatability and reliability of the experiment. In the PCA 3D map (Appendix A), the separation and aggregation of samples can be seen more intuitively. Three principal components were analyzed to find the difference of metabolites between and within groups.

### 3.4. Orthogonal Projections to Latent Structure-Discriminant Analysis (OPLS-DA)

The principal component analysis (PCA) described above is effective in extracting major information, but it is insensitive to variables with small correlation, which can be solved by partial least squares-discriminant analysis (PLS-DA). PLS-DA is a multivariate statistical analysis method with supervised pattern recognition. Specifically, the components in the independent variable X and dependent variable Y are extracted, respectively, and then the correlation between the components is calculated. Orthogonal projections to latent structure-discriminant analysis (OPLS-DA) [24] combine the orthogonal signal correction (OSC) and PLS-DA methods, which can decompose the information of the X matrix into two kinds of information related to Y and unrelated to Y, and screen the difference variables by removing the unrelated differences. The metabolome data were analyzed according to the OPLS-DA model, and scores of each group (Appendix A) were drawn to further display the differences among each group. Significant differences between the groups could be seen. Figure 4 is the verification of the above model. R^2^X, R^2^Y, and Q^2^ are the prediction parameters of the evaluation model, where R^2^X and R^2^Y represent the interpretation rate of the established model to the X and Y matrix, respectively, and Q^2^ represents the prediction ability of the model. The closer these three values are to 1, the more stable and reliable the model. When Q^2^ is greater than 0.5, it can be considered an effective model, and when Q^2^ is greater than 0.9, it is an excellent model. The validation diagram of the OPLS-DA model between different comparison groups is shown in Figure 4. One can clearly see the difference between AJH and MG (R^2^X = 0.919, R^2^Y = 1, Q^2^ = 0.998), between AJH and WH (R^2^X = 0.889, R^2^Y = 1, Q^2^ = 0.999), between AJH and YD (R^2^X = 0.929, R^2^Y = 1, Q^2^ = 0.999), and between AJH and GD (R^2^X = 0.918, R^2^Y = 1, Q^2^ = 0.998). These results demonstrate the stability and reliability of the models. Thus, the models could be used for further screening and identification of flavonoid metabolites.

### 3.5. Screening and Kegg Analysis of Flavonoid Differential Metabolites

Based on the results of OPLS-DA, flavonoid metabolites for each comparison group were screened by combining fold change as well as variable importance in project (VIP) values. In the study, the comparison of metabolites among five varieties identified 132 differential flavonoid metabolites. The difference in the expression level of metabolites in two group samples and the statistical significance of the difference was quickly identified by volcano plot (Figure 5A–D). The Venn diagram (Figure 5E,F) shows the relationship between different metabolites in each comparison group. A total of 88 significantly different flavonoid metabolites were screened between AJH and MG (43 less accumulated, 45 more accumulated), 90 between AJH and WH (40 less accumulated, 50 more accumulated), 88 between AJH and YD (less accumulated, 51 more accumulated), and 98 between AJH and GD (32 less accumulated, 66 more accumulated). In addition, 68 common differential metabolites were observed in the Venn diagram (Figure 5E) where each comparison group intersected, and each comparison group had its own unique differential metabolites. Therefore, differential metabolites could clearly distinguish AJH from other watercress varieties. Furthermore, we took an intersection of each comparison group among four watercress varieties without AJH in a Venn diagram (Appendix A). The differences in flavonoid metabolites made the four different watercress varieties distinguishable from each other.

The differential flavonoid metabolites between the five cultivars were mapped to the KEGG database. The KEGG classification results and enrichment analysis (Figure 6A–D) showed that the different flavonoid metabolites in the comparison group were involved in metabolic pathways, flavonoid biosynthesis, flavone and flavonol biosynthesis, biosynthesis of secondary metabolites, and anthocyanin biosynthesis.

## 4. Discussion

Past studies have shown that isoflavones are mostly found in legumes, such as soybeans. Consuming isoflavones can reduce the risks of breast cancer in women [25]. In addition, soy isoflavones are associated with bone in the human body and prevent osteoporosis-related bone loss [26]. In the present study, three isoflavones were detected in watercress and non-heading Chinese cabbage, all of which are genistein and its derivatives. Genistein helps prevent many chronic diseases (including solid tumors) by inhibiting new blood vessels [27]. Furthermore, genistein has a variety of inhibitory effects on breast cancer and can be used as an anticancer drug with great application prospects [28]. We found that there was only one isoflavone in AJH, while the GD had two isoflavones with the most varieties. Moreover, 22 flavones were detected in all the samples. Further, there are most kinds of flavones in GD. As one of the flavones, 6-C-MethylKaempferol-3-glucoside had not been studied before. Anthocyanin is a kind of natural pigment widely existing in plants and is related to the color of plants. In addition, studies have shown that anthocyanins are the main antioxidant activity contributors to protection of plants from adverse environmental impacts [29,30]. A total of eight anthocyanins were found in the tested samples. Interestingly, all anthocyanin levels in watercress were lower than those in AJH. In the present study, it was found that only one cyanidin-based anthocyanin was found in the MG group, while four cyanidin-based anthocyanins and two delphinium-based anthocyanins were found in the GD group. Moreover, anthocyanin was not detected in the WH and YD group. In addition, there are two anthocyanins that have not been reported in previous studies, namely delphinidin-3,5,3′-Tri-O-glucoside and cyanidin-3-O-(6″-O-p-coumaroyl) sophoroside-7-O-glucoside.

Flavanols, as bioactive compounds, are found in cocoa, red wine, green tea, red grapes, berries, and apples. Flavanols are also powerful antioxidants that scavenge free radicals both in vivo and in vitro. In this report, flavanol (naringenin-7-O-glucoside) was detected in watercress, and Han et al. found that naringenin-7-O-glucoside had a protective effect on oxidative stress of H9c2 cardiomyocytes induced by Adriamycin [31]. Eleven flavonoid carbonosides were detected in the study. Six flavonoid carbonosides were detected in the seeds of *E. ferox* [32]. The accumulation of flavonoid and flavonoid carbonoside in yellow passion fruit was significantly higher than that in purple fruit [33]. There are not many other studies on flavonoid carbonoside. Flavonols are a specific class of phenolics that are widely distributed in plants, where they function as antioxidants, antimi, crobials, photoreceptors, visual attractors, feeding repellants, and light screeners. 3 dihydroflavonol were detected in this study, one of which (Hesperetin-5-O-glucoside) had not been reported in previous studies.

A total of 132 flavonoid metabolites were detected in this experiment. There were six unique metabolites in AJH (Figure 5F). They were Genistein-7-O-galactoside, Naringenin-7-O-glucoside (Prunin), Luteolin-6-C-glucoside-7-O-glucoside, Luteolin-6,8-di-C-glucoside, Cyanidin-3-O-(6″-O-p-coumaroyl) sophoroside-7-O-glucoside, and Kaempferol-3-O-sophorotrioside-7-O-glucoside. Studies have shown that naringenin-7-O-glucoside has a protective effect on doxorubicin-induced apoptosis, and may be helpful for the treatment or prevention of doxorubicin-related cardiomyopathy [31]. There were 13 unique metabolites in GD (Figure 5F). They were Kaempferide (3,5,7-Trihydroxy-4′-methoxyflavone), Rhamnetin, Azaleatin (5-O-Methylquercetin), Chrysoeriol-7-O-glucoside, 6-C-Methyl Kaempferol-3-glucoside, Apigenin-6-C-(2″-xylosyl)glucoside, Isosaponarin (Isovitexin-4′-O-glucoside), Quercetin-3-O-xylosyl(1→2)glucoside, Quercetin-3-O-(2″-O-rhamnosyl) galactoside, 2′-Hydoxy,5-methoxyGenistein-4′,7-O-diglucoside, Luteolin-6-C-glucoside-7-O-(6″-p-coumaroyl)glucoside, Quercetin-3-O-rutinoside-7-O-glucoside, and Luteolin-6-C-glucoside-7-O-(6″-feruloyl)glucoside. As a kind of natural flavonoid, Kaempferide (3,5,7-Trihydroxy-4′-methoxyflavone) has strong anticancer activity in many human tumor cells [34]. Studies have shown that Rhamnetin can be used as a promising radiosensitizer to improve the efficacy of radiotherapy in humans [35]. Quercetin-3-O-rutinoside-7-O-glucoside was found in *L. chinense* leaves, and it has potential applications in the nutraceutical field [36]. Metabolites unique to GD variety may play a greater role in human health.

The UHPLC-ESI-MS/MS method used in this experiment cannot accurately distinguish isomers. All isomers are indicated by asterisks in Appendix A. Isomers are shown because the signal of substances can be detected by mass spectrometry and such substances exist. The substance information detected matches the substance information in the database, but it is not clear which isomer it is. Mass spectrometry is not able to determine the structure of a compound. Metabonomics based on mass spectrometry is of significance in high sensitivity and wide screening.

## 5. Conclusions

In this study, the UHPLC-ESI-MS/MS metabolomics method was used to evaluate the difference of metabolites between non-heading Chinese cabbage and watercress. This is the first attempt to study the metabolomics between non-heading Chinese cabbage and watercress and between different varieties of watercress. A total of 132 flavonoid metabolites (including 8 anthocyanins, 2 dihydroflavone, 3 dihydroflavonol, 1 flavanols, 22 flavones, 11 flavonoid carbonosides, 82 flavonols, and 3 isoflavones) were detected. Flavonoid metabolites varied widely in different samples. Both the non-heading Chinese cabbage and the variety of watercress from Guangdong, China, had their own unique metabolites.

## Figures and Tables

**Figure 1 molecules-26-05825-f001:**
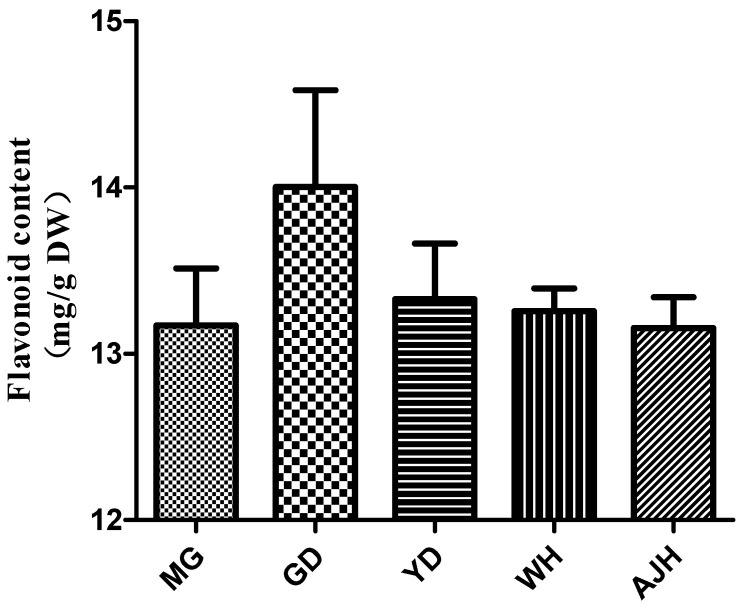
Total flavonoid contents of all the samples in the experiment.

**Figure 2 molecules-26-05825-f002:**
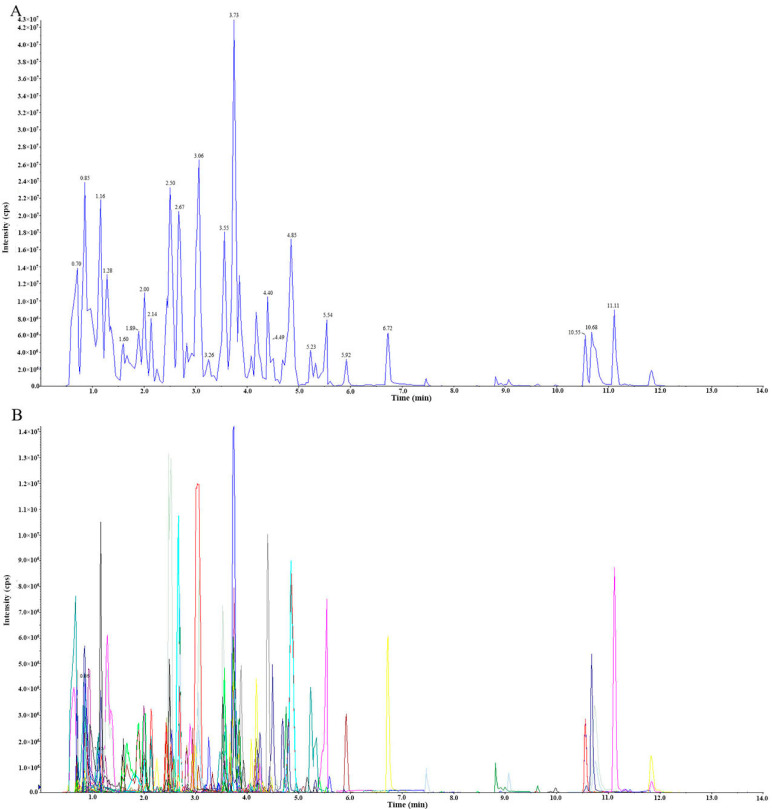
The TIC of mixed QC sample by mass spectrometry detection (**A**) and multi-peak detection plot of metabolites in the MRM mode (**B**).

**Figure 3 molecules-26-05825-f003:**
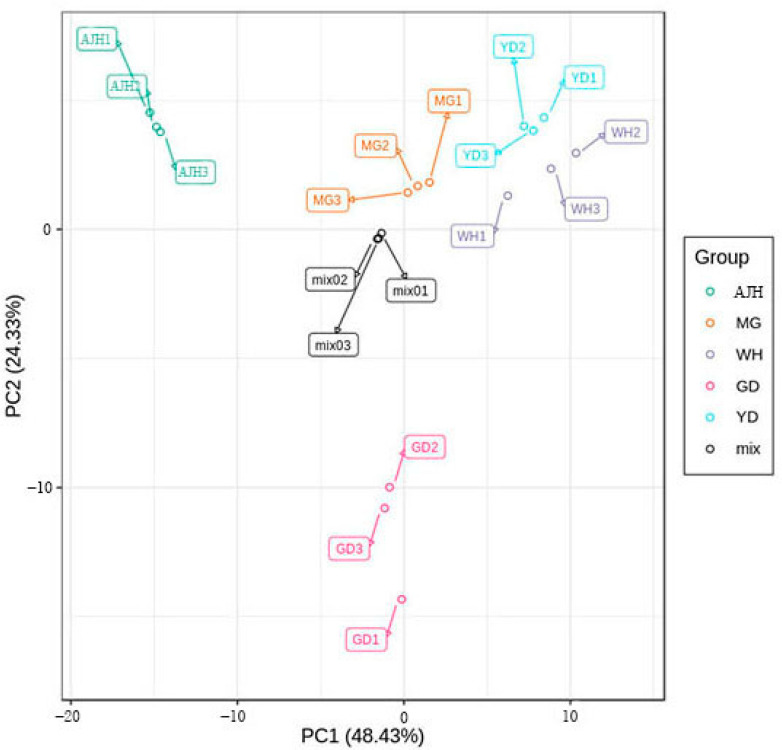
The principal component (PCA) score plot of each group sample and quality control sample.

**Figure 4 molecules-26-05825-f004:**
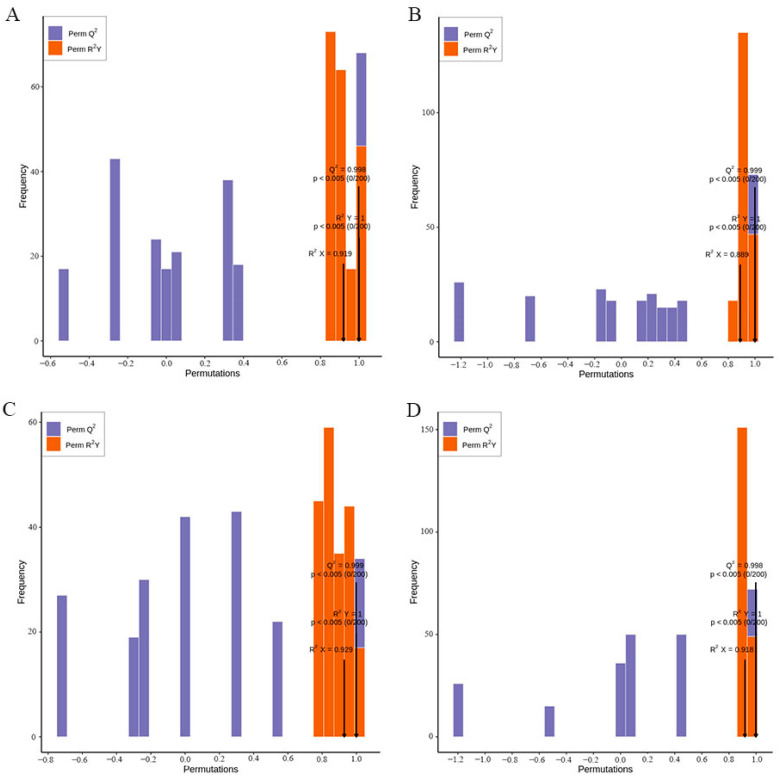
Permutation test of the OPLS-DA model for the comparison group of AJH versus MG (**A**), AJH versus WH (**B**), AJH versus YD (**C**), and AJH versus GD (**D**).

**Figure 5 molecules-26-05825-f005:**
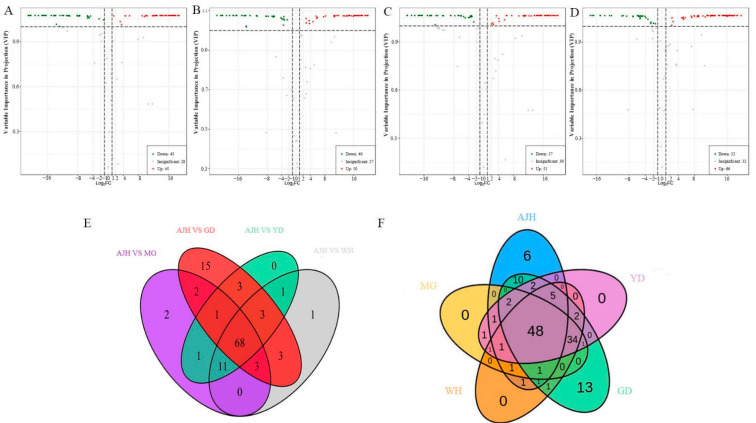
Volcanic plots and Venn diagram of differential metabolites for each comparison group. (**A**–**D**) The volcanic plots of the comparison group AJH versus MG, AJH versus WH, AJH versus YD, and AJH versus GD, respectively; (**E**) Venn diagram showing the overlapping and unique differential metabolites among the comparison groups; (**F**) Venn diagram showing the overlapping and unique metabolites among the comparison.

**Figure 6 molecules-26-05825-f006:**
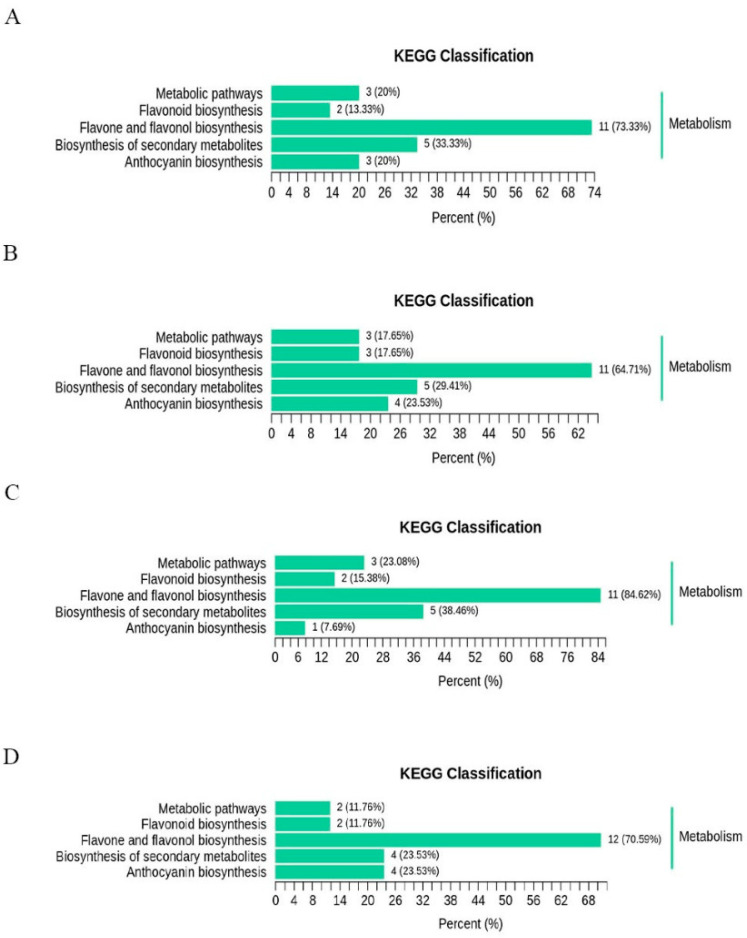
The differential metabolites KEGG classification of the comparison group AJH versus MG (**A**), AJH versus WH (**B**), AJH versus YD (**C**), and AJH versus GD (**D**), respectively.

**Table 1 molecules-26-05825-t001:** Analytical conditions of UHPLC.

UHPLC Conditions	Parameters	
The chromatographic columns	Agilent SB-C18 1.8 µm, 2.1 mm × 100 mm	
Mobile phase	Phase A: ultra-pure water (0.1% formic acid was added) phase B: acetonitrile (0.1% formic acid was added)	
Gradient program	0 min	95:5 (*v*:*v*)
	9.0 min	5:95 (*v*:*v*)
	10.0 min	5:95 (*v*:*v*)
	11.0 min	95:5 (*v*:*v*)
	14.0 min	95:5 (*v*:*v*)
Flow rate	0.40 mL/min	
Temperature	40 °C	
Injection volume	5 μL	

**Table 2 molecules-26-05825-t002:** Analytical conditions of mass spectrometry.

Mass Spectrometry Condition	Parameters	
Ion source	Turbo spray	
Source temperature	550 °C	
Ion spray voltage	5500 V	
Ion source	Gas I	55 psi
	Gas II	60 psi
	Curtain gas	25 psi
Collision gas	High	

## Data Availability

The data presented in this study are available on request from the corresponding author. The data are not publicly available because some other results have not been analyzed and published publicly and these results cannot be isolated.

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
