# Peer review of "Analysis of Flavonoid Metabolites in Watercress (Nasturtium officinale R. Br.) and the Non-Heading Chinese Cabbage (Brassica rapa ssp. chinensis cv. Aijiaohuang) Using UHPLC-ESI-MS/MS"

_molecules, 2021, doi:10.3390/molecules26195825_

Round 1

Reviewer 1 Report

Authors should extend the introduction and emphasize the novelty of their study. Also, they should put more details on UHPLC method. 

In addition, the authors use the abbreviation UPLC, which is only for Waters instruments and consumables. Please correct it to UHPLC. 

Reviewer 2 Report

The manuscript “Analysis of Flavonoid Metabolites in watercress (Nasturtium officinale R. Br.) and the non-heading Chinese cabbage (Brassica rapa ssp. chinensis cv. Aijiaohuang) Using UPLC-ESI-MS/MS” was submitted to Molecules for publication.

Broad comments:

The authors compare the metabolite pattern of watercress to the non-heading Chinese cabbage, also known as pak choi. The reason for the study is that the two mentioned species display the highest nutrient density score according to Di Noia (2014). The experimental part of the work at first glance seems very well performed and solid but still reveals several questions.

The first question as in so many of these kind of studies is the identity of the compounds. The authors say that a 132 two components were identified but when looking at table S1 the majority of the compounds display an asterisk indicating indistinguishable isomers. What does mean isomer then? Substituted at another position or another type of sugar? The authors at the end of the manuscript admit that with LC-MS it is not possible to determine the structure of a compound (line 258); so how they can be sure that they identified correctly?

Another point is the quantitation. Doing MRM with 132 compounds would mean that the 132 compounds would have been available as standards, but this does not seem very likely as the authors are not even sure which components they are dealing with.

Apart from the technical shortcomings the scientific questions is: “what does one expect when five plant samples are compared of which one is not even from the same genus?” Of course, the one species will give most divers results, which is still ok when the one species is treated as an outliner. But the authors, in contrast were leveling pak choi against each of the watercress species instead of leveling intraspecifically.

In the discussion section the authors do try to draw some relations of detected compounds to reported literature on beneficial effects. They start with isoflavones of which they claim that they detected three compounds that were all genistein, which is again not possible because only one compound at one time can be genistein, while the other two are derivatives thereof. Apart from that, the impact of genistein in cruciferous vegetables has to be questioned as there exist vegetables with manifold higher concentrations. Later on the authors connect to the passion fruit and subsequently pick out some of the compounds randomly, to have something to write on.

Summarizing, the present work does not only show methodical inconsistencies but also has no substance at all, which becomes most evident in the discussion part. Here it would make much more sense to analyze the main compounds and quantify their real content in the mentioned species. Hence, it would at least be reasonable to attribute certain health-beneficial effects to the respective foods.

Specific comments:

Table S1: Why does heptamethoxyflavone appear in the middle of glycosides though it is by far more apolar than any of the other components?

Reviewer 3 Report

The paper is based on evaluation of flavonoids profile in watercress and Chinese cabbage.

More information about the samples must be provided: when and where collected (GPS coordinated maybe included).

The procedure for evaluation of flavonoids content is not totally clear. Usually some reference compounds as equivalents are used (rutin or catechin). Was something in your study?

List of reagents with purities and manufacturers must be added.

There is no Figure S2.

Explain (name) the main peaks in Figure 2.

Check latin names in the manuscript, they should be in italic. 

In my opinion, conclusions should be written in separate section (check journal requirements). Add more details in conclusions, as now they are very abstract.

Round 2

Reviewer 2 Report

The authors gave some explantations in their response but did not significantly modify their mansuscript or adapt the design of the study. Thus, I still don't see the study to have the necessary quality for publication in Molecules.